# Effects of Preoperative Oral Nutritional Supplements on Improving Postoperative Early Enteral Feeding Intolerance and Short-Term Prognosis for Gastric Cancer: A Prospective, Single-Center, Single-Blind, Randomized Controlled Trial

**DOI:** 10.3390/nu14071472

**Published:** 2022-04-01

**Authors:** Feng-Jun He, Mo-Jin Wang, Kun Yang, Xiao-Long Chen, Tao Jin, Li-Li Zhu, Wen Zhuang

**Affiliations:** 1West China School of Medicine, West China Hospital, Sichuan University, Chengdu 610044, China; He2021324025259@163.com (F.-J.H.); taojin_068@163.com (T.J.); zhulili_1121@163.com (L.-L.Z.); 2Department of Gastrointestinal Surgery, West China Hospital, Sichuan University, Chengdu 610044, China; wangmojin2001@163.com (M.-J.W.); yangkun068@163.com (K.Y.); xiaolong_chen1988@163.com (X.-L.C.)

**Keywords:** feeding intolerance, enhanced recovery after surgery, enteral nutrition, gastric cancer, oral nutritional supplements, randomized controlled trial, gastroenterostomy

## Abstract

Background: Early enteral nutrition (EN) after abdominal surgery can improve the prognosis of patients. However, the high feeding intolerance (FI) rate is the primary factor impeding postoperative EN. Methods: Sixty-seven patients who underwent radical subtotal or total gastrectomy for gastric cancer (GC) were randomly allocated to the preoperative oral nutritional supplement group (ONS group) or dietary advice alone (DA group). Both groups were fed via nasojejunal tubes (NJs) from the first day after surgery to the fifth day. The primary endpoint is the FI rate. Results: Of the patients, 66 completed the trial (31 in the ONS group, 35 in the DA group). The FI rate in the ONS group was lower than that in the DA group (25.8% vs. 31.4%, *p* = 0.249). The postoperative five-day 50% energy compliance rate in the ONS group was higher than that in the DA group (54.8% vs. 48.6%, *p* = 0.465). The main gastrointestinal intolerance symptoms were distension (ONS vs. DA: 45.2% vs. 62.9, *p* = 0.150) and abdominal pain (ONS vs. DA: 29.0% vs. 45.7%, *p* = 0.226). Postoperative nausea/vomiting rate and heartburn/reflux rate were similar between the two groups. We noted no difference in perioperative serum indices, short-term prognosis or postoperative complication rates between the two groups. Conclusions: The study shows that short-term preoperative ONS cannot significantly improve FI and the energy compliance rate in the early stage after radical gastrectomy.

## 1. Introduction

Early EN after major abdominal surgery is more beneficial to improving prognosis than total parenteral nutrition (TPN) [1], while not everyone receives early EN due to the high incidence of intolerance symptoms [2]. FI may be as high as 75% in ICU inpatients, 49.3~68% in patients after radical gastrectomy, and possibly with poor prognosis [3,4]. FI was found to be an independent risk factor for postoperative complications in patients with colorectal cancer after colorectal surgery and mostly occurred on the third day after the start of EN treatment [5,6]. A retrospective study [7] found that the presence of more than two types of gastrointestinal peristalsis symptoms is associated with a longer postoperative hospital stay, readmission, increased postoperative infection complications and mortality, but the strength of this relationship may depend on the definition of FI used [3].

Appropriate feeding routes, paying attention to oral hygiene, proper positioning for tube feeding (30–45° semidecubitus position), adopting nurse-led management of EN and selecting appropriate nutrition can variably improve postoperative FI [8]. In addition, the influence of cultural practices, religious practices and dietary culture cannot be ignored [9]. The mechanism by which preoperative ONS modulates tolerance to early EN after gastrointestinal surgery remains unclear. Animal experiments showed that food stimulation played a regulatory role in the mouse intestinal microecology, while alternately, accepting a high-fat, high-sugar diet or a low-fat, high-plant polysaccharide diet can lead to a periodically dynamic adjustment in the mouse intestinal microecological system, and each intestinal flora fluctuation caused by specific food reaches a new equilibrium after 3.5 days. Most of the changes in flora characteristics are reversible [10]. Numerous studies effectively improved gastrointestinal symptoms such as food component intolerance and inflammatory bowel disease through targeted induction and regulation of intestinal flora by dietary components [11,12].

Food is a crucial environmental factor regulating intestinal flora to reduce symptoms of food intolerance. Our study is the first randomized controlled study to explore ways to improve FI after radical gastroenterostomy through preventive intervention.

## 2. Materials and Methods

### 2.1. Study Design

The single-center, two-arm, parallel-group, randomized controlled trial was conducted at the Department of Gastrointestinal Surgery, West China Hospital, Sichuan University, from June 2020 to February 2021 approved by the Biomedical Ethics Committee of West China Hospital of Sichuan University [2018 Review (468)] and registered in the ChiCTR under ChiCTR2000034961.

### 2.2. Patients

Patients treated in the Center of Gastrointestinal Surgery, West China Hospital, Sichuan University, were eligible if they were between 18 and 80 years of age, were to undergo elective radical gastrectomy for histologically confirmed adenocarcinoma and were diagnosed at the clinical stage of T2-4aN0-3M0 according to the Japanese Classification [13]. Detailed eligibility criteria are shown in Table 1.

### 2.3. Randomization and Blinding

Eligible patients were randomly allocated in a 1:1 ratio to the ONS group or DA group. Randomization sequences were generated by the trial’s assistant with the use of random-number tables. Since this study involved home enteral nutrition (HEN), patients could not be blinded, but clinicians and nurses remained blind to the allocated group of each participant until the data analysis was completed.

### 2.4. Procedures

The ONS group began preoperative ONS treatment with Ruidai (TPFD, 500 mL/bag, Fresenius Kabi Deutschland GmbH, Bad Homburg, Germany) for seven days, which contained 0.9 kcal/mL. The expected daily nutritional intake in the ONS group was 450 kcal energy, 17 g protein, 16 g fat and 60 g carbohydrate per ONS pack. The control group was given dietary advice. The amount of daily ONS intake was recorded and checked by the dietitian for the ONS group during the consultation sessions.

During the perioperative period, all patients were treated according to the principles of the ERAS protocols. Radical gastrectomy (R0 resection) and D2 lymph node dissection refer to the principles of New Japanese classifications and treatment guidelines for gastric cancer by the Japanese Gastric Cancer Association (JGCA) were performed by professors with rich clinical experience [14]. A nasojejunal tube (Freka^®^Tube CH/FR15 120 cm) was placed in the efferent loop approximately 20 cm behind the anastomosis after digestive tract reconstruction.

EN was started on the first day postoperatively by a nasojejunal tube. On the first day, Ruidai was pumped continuously at an initial rate of 20 mL/h through an enteral feeding pump (APPLIX^®^SMART, Fresenius Kabi AG, Bad Homburg, Germany) and increased by 20 mL/h per day until the fifth day after surgery. At the same time, they gradually recover from liquid–semiliquid–soft food and increase oral intake.

### 2.5. Outcomes

The primary endpoint of this study was the incidence of FI and to evaluate the effect of preoperative ONS on postoperative FI (FI: Tube feeding is interrupted for more than 24 h, FI rate = number of FI cases (*n*)/sample size (*n*) × 100%). The secondary endpoint was the rate of energy supply by EN up to 50% of the target daily energy requirement (25 kcal/kg/d) within 5 days after surgery.

The index in our study included postoperative gastrointestinal symptom rate, serum prealbumin (PAB), total protein (TP), albumin (ALB), lymphocyte count, procalcitonin (PCT), C-reactive protein (CRP) and interleukin-6 (IL-6) levels after surgery; postoperative complication rate (complication classification was based on Clavien-Dindo grading standard (20)); and one-month unplanned readmission rate.

Gastrointestinal intolerance: occurrence of any of the following gastrointestinal symptoms and symptom score ≥ 3 (abdominal distention, abdominal pain, nausea/vomiting, heartburn/reflux, diarrhea).

### 2.6. Statistical Analysis

Prospective studies are lacking. The sample size was calculated based on previous studies [4]. The expected rate of FI was estimated at 20%, and its threshold was estimated at 58%, averaged from existing studies. With a statistical power of 90% and a one-sided type I error of 5%, the number of eligible patients required for this study was calculated to be 60 by NCSS PASS 11. By considering the 10% exit status, the expected sample size was 66 patients. Categorical variables were compared using Pearson’s chi-squared test and Fisher’s exact tests. Nonparametric continuous variables were compared using Mann–Whitney *U* tests and reported as medians and standard deviations. Statistical Package for Social Science version 26 (SPSS Inc., Chicago, IL, USA) was used for all statistical analyses.

## 3. Results

A flowchart of the study is depicted in Figure 1. Sixty-seven patients were enrolled in this study (32 experimental group and 35 control subjects), and one participant in the experimental group was removed from the study due to abandonment of the operation. The clinical backgrounds and surgical findings of the patients are shown in Table 1. There were no significant differences in sex, median age, median weight loss, education, Nutrition Risk Screening 2002, body mass index (BMI), underlying diseases, tumor site, operation time, surgical approach or American Society of Anesthesiologists between the two groups.

The findings of early tube feeding are shown in Table 2. There were no significant differences in the incidence of FI (25.8% vs. 31.4%, *p* = 0.615), the 50% energy compliance rate on POD 5 (54.8% vs. 48.6%, *p* = 0.611) or the postoperative average tube feeding amount (2260 ± 982 mL vs. 2365 ± 934 mL, *p*= 0.657). The incidence of postoperative gastrointestinal symptoms (GIS) in the ONS group and DA group was 58.1% and 68.6% (*p* = 0.436), respectively, without statistical significance. The major intolerance symptoms were abdominal distention (ONS vs. DA: 45.2% vs. 62.9%, *p* =0.150) and abdominal pain (ONS vs. DA: 29% vs. 45.7%, *p* =0.163). Enteral feeding intolerance mainly occurred approximately 3 days after surgery (ONS vs. DA: 2.95 ± 1 vs. 2.93 ± 0.8, *p* = 0.943). There was no significant difference in the time to first flatus between the two groups. Detailed eligibility criteria are shown in Figure 2.

Changes in proteins and inflammatory indicators during the perioperative period are shown in Figure 3. Albumin levels tended to be higher in the DA group than in the ONS group on POD 1 (*p* = 0.006). There were no significant differences in prealbumin (PAB), total protein (TP), lymphocytes, procalcitonin (PCT), C-reactive protein (CRP), interleukin-6 (IL-6) or blood glucose levels between the two groups.

Morbidities are shown in Table 3. In the ONS group, two (6.5%) developed a pulmonary infection, in addition to 4 of 35 patients (11.4%) in the DA group. One developed anastomotic fistula in the ONS. There was no significant difference in the unplanned readmission rate between the two groups within 1 month after discharge. 

The results of the subgroup analysis are shown in Table 4. There was no baseline difference between the two subgroups. In the ONS group, the incidence of FI after distal subtotal gastrectomy (DG) was significantly higher than that after total gastrectomy (TG) (41.2% vs. 8.3% *p* = 0.026), and the incidence of abdominal distension in the TG was significantly lower than that in the DG (*p* = 0.010). In the DA group, the incidence of FI after DG also tended to be higher than that after TG (33.3% vs. 20%, *p* = 0.961). In the DA group, the incidence of hiccups after TG was higher than that after DG (TG vs. DG: 0 vs. 20%, *p* = 0.024). The FI rate after TG in the ONS group was lower than that after TG in the DA group (ONS vs. DA: 8.3% vs. 30.0%, *p* = 0.190).

## 4. Discussion

The definition of FI is controversial. It is currently diagnosed in the following three ways: gastric residual volume (GRV), subjective symptoms of gastrointestinal discomfort and insufficient enteral feeding. Jean Reignier et al. [15] found that routine measurement of RGV failed to improve the prognosis but hindered the progression of EN. Therefore, the ASPEN guidelines stress that the diagnosis of FI should focus more on subjective symptoms than routine GRV measurement [16].

Symptoms are difficult to evaluate because they can usually be described in multiple ways. If evaluating one aspect at a time, the accuracy of the assessment may be improved [17]. Our study adopted a more simplified autonomous gastrointestinal symptom assessment form based on the form used by Svolos V et al. [18] and took the tube feeding energy supply as an objective indicator to assess tolerance to EN.

The results showed that the incidence of early postoperative FI was 25.8% in the ONS group and 31.4% in the control group. The 7-day ONS before operation failed to reduce the early FI rate and increase the EN energy supply rate. The incidence of postoperative GIS was 58.1% in the ONS group and 68.6% in the DA group, which is consistent with the 49.3%~68% reported in previous studies [3,4]. Our study found that abdominal distension, abdominal pain, nausea/vomiting, heartburn and reflux were the most common intolerance symptoms of early EN after radical gastric cancer surgery, and the incidence of diarrhea was lower, which was similar to the results of the meta-analysis [19]. In this study, 86.1% of patients were at risk of malnutrition, which is consistent with previous investigations.

The mechanism of FI is still unclear. After gastrectomy, the physiological structure of the digestive tract changes, and hormone levels and peristaltic function change [20,21]. FI is mainly caused by the decrease in gastrointestinal peristalsis after surgery [22]. The detection of serum concentrations of gastrointestinal hormones may be an important way to evaluate gastrointestinal function, but there is still no one that is generally used to monitor postoperative gastrointestinal function effectively [23]. Disease burden, surgical trauma, anesthesia, postoperative systemic inflammatory response, intestinal flora imbalance and underlying diseases may all be contributors to early FI [24,25]. A retrospective study found that the grade of anesthesia, the level of inflammation and the pain score within 6 h after surgery are independent risk factors for the occurrence of early postoperative FI (4). Statistics show that the incidence of gastroparesis after abdominal surgery is approximately 19%, among which the rate of diabetes can reach 30%–50% [26]. Food intolerance also exists in approximately 20% of the population. Researchers try to find ways to treat food intolerance by using the food itself. Therefore, food is expected to become a good medicine for certain chronic diseases in humans, such as diabetes [27], as is the intestinal flora regulated by food [28]. For example, inulin-type fructans were used to target the intestinal flora, thereby activating the nitric oxide (NO) synthase/NO pathway and reversing endothelial dysfunction in mesenteric and carotid arteries of n-3 PUFA-depleted Apoe-/- mice [29].

Our study found that early FI mostly occurred 3 days after the operation, which is consistent with that reported in a retrospective study [4]. At the early stage after surgery, with a severe systemic inflammatory response, the purpose of early EN may not be just blindly oriented to achieve energy goals [30]. Furthermore, early EN can alleviate the inflammatory response, nourish the intestinal mucosa, prevent bacterial translocation and reduce postoperative infection complications when the energy supply reaches 10–20% of the body’s energy demand [31].

We found that the incidence of GIS after total gastrectomy was lower than that after distal subtotal gastrectomy. At present, there is a lack of relevant research reports. Our study also showed that the incidence of hiccups after TG was higher than that after DG, which may be related to the mechanism of Roux-Y stasis syndrome, but the significance of different methods of digestive tract reconstruction and the relationship with long-term quality of life still needs further exploration.

This study is a prospective randomized controlled study that strictly follows the CONSORT statement for quality control [32] but is limited by the small sample size. Second, the study did not further investigate the mechanism. It was not considered that different digestive tract reconstruction methods had different effects on early FI and short-term quality of life.

## 5. Conclusions

Further research is needed to identify the effect of preoperative ONS on tolerance to EN and the energy compliance rate in the early stage after radical gastrectomy.

## Figures and Tables

**Figure 1 nutrients-14-01472-f001:**
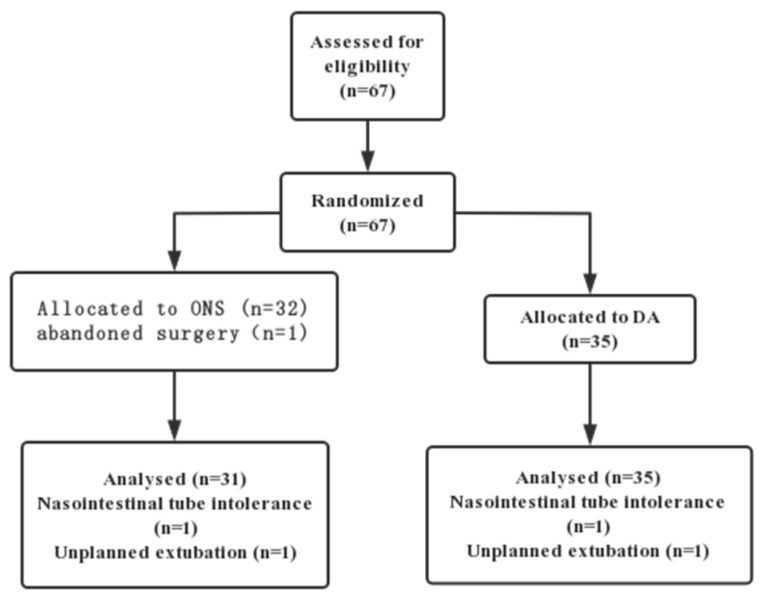
CONSORT flow diagram of patient recruitment and randomization.

**Figure 2 nutrients-14-01472-f002:**
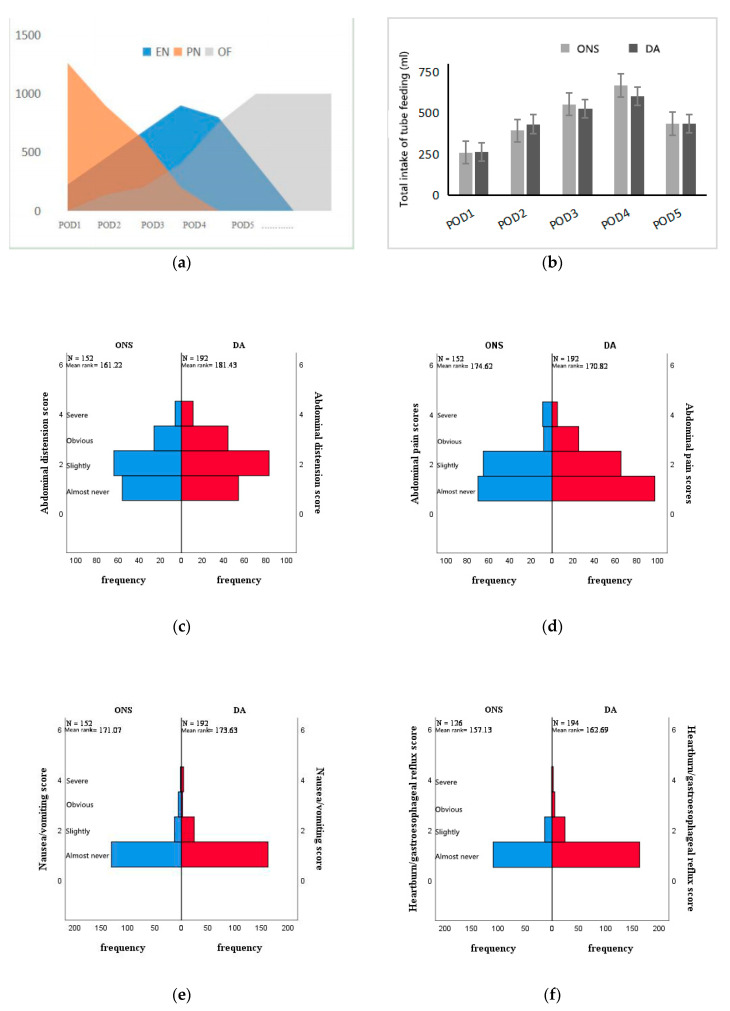
(**a**) Patterns from parenteral nutrition to oral overfeeding. (**b**) Average daily tube feeding amount of experimental group and control group after operation. (**c**–**h**) Abdominal distension, abdominal pain, nausea/vomiting and heartburn/reflux symptom scores were compared between experimental group and control group after operation. (**g**): Proportion of each symptom score in the total scores of gastrointestinal symptoms after surgery. (**h**): Postoperative mean daily total score of gastrointestinal symptoms.

**Figure 3 nutrients-14-01472-f003:**
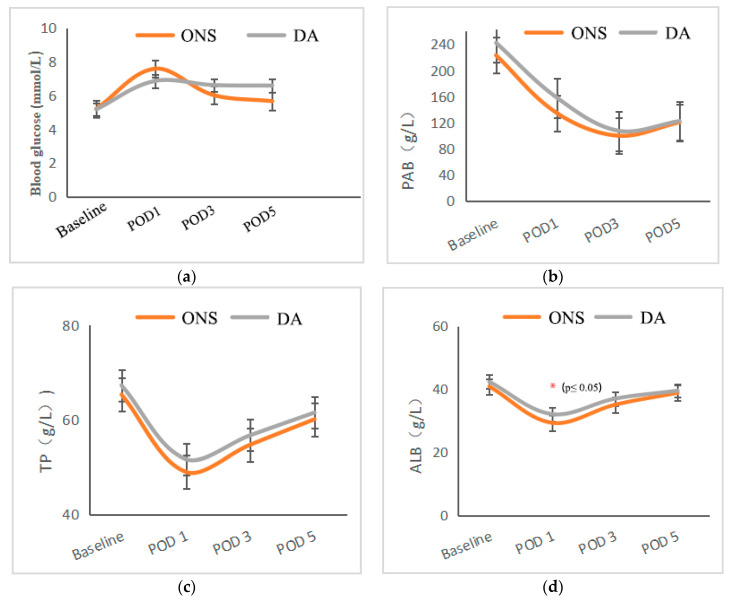
Changes in serological indicators. (**a**) Perioperative blood glucose fluctuation. (**b**) Perioperative prealbumin (PAB) fluctuation. (**c**) Perioperative fluctuation of serum total albumin (TB). (**d**) Perioperative serum albumin (ALB) fluctuation. (**e**) Perioperative c-reactive protein fluctuation (CRP). (**f**) Perioperative interleukin-6 (IL-6) fluctuation. (**g**) Changes in serum procalcitonin (PCT) levels during perioperative period. (**h**) Perioperative white blood cell (WBC) fluctuation. (**i**) Changes in Serum Lymphocyte Count.

**Table 1 nutrients-14-01472-t001:** Patient characteristics.

Patient Characteristics	ONS (*n* = 31)	DA (*n* = 35)	*p*
Sex (Male vs Female)	24 vs. 7	23 vs. 12	0.295
Age ^∆^ (y)	63.2 ± 12.0	60.5 ± 9.4	0.266
Weight loss ^∆^ (Kg)	3.5 ± 3.1	2.4 ± 2.9	0.07
BMI ^∆^ (Kg/m^2^)	22.00 ± 3.0	23.13 ± 2.4	0.087
NRS2002 Score ^∆^	3.55 ± 1	3.44 ± 1	0.709
ONS days	7.6	-	-
Mean total oral intake (mL)	3822.58	-	-
Mean daily oral intake (Kcal)	452.67	-	-
Hypertension (*n*, %)	3 (9.7)	9 (24.3)	0.092
Diabetes mellitus (*n*, %)	3 (9.7)	9 (24.3)	0.092
Smoke (*n*, %)	8 (25.8)	9 (24.3)	0.499
Drink (*n*, %)	9 (29.0)	11 (29.7)	0.967
Tumor location (*n*, %)			0.771
Esophagogastric junction	5 (16.1)	5 (14.3)	
Gastric body	11 (35.5)	10 (28.6)	
Antrum	15 (48.4)	20 (57.1)	
Meanmaximum diameter of tumor (cm)	4.7 ± 2.8	4.3 ± 2.6	0.389
Gastrectomy			0.375
Distal subtotal gastrectomy	17 (54.8)	24 (65.7)	
Proximal subtotal gastrectomy	2 (6.4)	1 (2.9)	
Total gastrectomy	12 (38.7)	10 (28.6)	
The operation time ^∆^ (min)	206.97 ± 33.1	203.30 ± 41.1	0.516
Intraoperative infusion volume ^∆^ (mL)	1959.68 ± 411.2	2021.05 ± 537.2	0.619
ASA			0.378
II	28 (90.0)	29 (82.9)	
III	3 (10.0)	6 (17.1)	
pTNM			0.57
IA	2	6	
IB	2	6	
IIA	4	3	
IIB	5	6	
IIIA	6	4	
IIIB	10	9	
IIIC	2	1	

^∆^: Mean ± standard deviation. BMI: Body Mass Index.

**Table 2 nutrients-14-01472-t002:** The study findings.

Enteral Nutrition and Feeding Intolerance Outcomes	ONS (*n* = 31)	DA (*n* = 35)	*p*
Feeding intolerance (*n*,%)	8 (25.8)	11 (31.4)	0.615
Severe gastrointestinal reactions (*n*,%)	6 (19.4)	8 (22.9)	0.366
Nasointesinal tube intolerance or unplanned extubation (*n*,%)	2 (6.5)	3 (8.6)	0.886
Symptoms of feeding intolerance (*n*,%)			
Abdominal distension (*n*,%)	14 (45.2)	22 (62.9)	0.150
Abdominal pain (*n*,%)	9 (29.0)	16 (45.7)	0.163
Nausea/vomiting (*n*,%)	7 (22.6)	4 (11.4)	0.225
Heartburn/gastroesophageal reflux (*n*,%)	3 (9.7)	5 (14.3)	0.567
Hiccup (*n*,%)	3 (9.7)	2 (5.7)	0.544
Diarrhea (*n*,%)	0 (0)	1 (2.9)	0.343
Incidence of symptoms of feeding intolerance (*n*,%)	18 (58.1)	24 (68.6)	0.436
Time of feeding decrement ^∆^ (POD days)	2.95 ± 1	2.93 ± 0.8	0.943
Anal exsufflation time ^∆^ (d)	3.1 ± 0.8	3.2 ± 0.7	0.839
Time of energy reaching standard ^∆^ (day)	3.59 ± 0.8	3.94 ± 0.8	0.214
50% energy compliance rate (%)	17 (54.8)	17 (48.6)	0.611
Total energy intake ^∆^ (Kcal)	2260 ± 982	2365 ± 934	0.657
Total protein intake ^∆^ (g/day)	58.58 ± 11.8	60.5 ± 9.5	0.450

^∆^: Mean ± standard deviation. Feeding intolerance: Interruption of enteral nutrient supply for more than 24 h. Severe gastrointestinal reactions: Gastrointestinal reactions leading to tube feeding interruption for more than 24 h. Symptoms of feeding intolerance: Any gastrointestinal symptoms such as abdominal distension/abdominal pain/nausea, vomiting/ heartburn, reflux/diarrhea with a scores greater than or equal to 3 points. Incidence of symptoms of feeding intolerance = Number of occurrence/Sample size.

**Table 3 nutrients-14-01472-t003:** Incidence of short-term complications and readmission rate.

Postoperative Complications	ONS (*n* = 31)	DA (*n* = 35)	*p*
Complications (*n*,%)	2 (6.5)	4 (11.4)	0.615
Pulmonary infection	0 (0)	1 (2.9)	
Gastroparesis	1 (3.2)	1 (2.9)	
Anastomotic fistula	1 (3.2)	0 (0)	
Allergy	0 (0)	1 (2.9)	
Posttraumatic stress disorder	0 (0)	1 (2.9)	
Unplanned readmission (%)	0 (0)	1 (0)	
Clavien-Dindo Classification			
II	2 (6.5)	4 (11.4)	

**Table 4 nutrients-14-01472-t004:** The incidence of gastrointestinal intolerance after distal subtotal gastrectomy and total gastrectomy in the experimental group.

ONS Group	DG	TG	*p*
Sex (Male vs Female)	14:3	9:3	0.630
Age ^∆^ (y)	61.6 ± 14.7	65.5 ± 8.3	0.421
BMI ^∆^ (Kg/m^2^)	22.6 ± 3.3	21.3 ± 2.7	0.312
ONS days (d)	7.8 ± 2.3	7.9 ± 2.8	0.924
NRS2002 Score ^Δ^	3.4 ± 1	3.7 ± 1	0.491
Hypertension (*n*, %)	3/17	1/12	0.124
Diabetes mellitus (*n*, %)	1/17	1/12	0.226
Smoke (*n*, %)	3/17	4/12	0.284
Drink (*n*, %)	5/17	4/12	0.822
Time of energy reaching standard ^Δ^ (day)	3.1 ± 0.7	4.0 ± 0.7	0.029 *
Time of feeding decrement ^Δ^ (days)	2.5 ± 0.9	3.17 ± 1.2	0.199
Symptoms of feeding intolerance (*n*,%)			
Abdominal distension (*n*,%)	11/17 (64.7)	2/12 (16.7)	0.010 *
Nausea/vomiting (*n*,%)	4/17 (23.5)	3/12 (25)	0.927
Heartburn/gastroesophageal reflux (*n*,%)	1/17 (5.9)	2/12 (16.7)	0.348
Hiccup (*n*,%)	2/17 (11.8)	1/12 (8.3)	0.765
Abdominal pain (*n*,%)	7/17 (41.2)	2/12 (16.7)	0.160
Feeding intolerance (*n*,%)	7/17 (41.2)	1/12 (8.3)	0.026 *
50% energy compliance rate(%)	7/17 (41.2)	9/12 (75)	0.071
Anus exhausting time	3.06 ± 0.6	3.08 ± 0.9	0.933

* *p* < 0.05, ^Δ^: Mean ± standard deviation, Feeding intolerance: Interruption of enteral nutrient supply for more than 24 h. Symptoms of feeding intolerance: Any gastrointestinal symptoms such as abdominal distension/abdominal pain/nausea, vomiting/ heartburn, reflux/ diarrhea with a scores greater than or equal to 3 points.

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
