# Peer review of "Effects of Preoperative Oral Nutritional Supplements on Improving Postoperative Early Enteral Feeding Intolerance and Short-Term Prognosis for Gastric Cancer: A Prospective, Single-Center, Single-Blind, Randomized Controlled Trial"

_nutrients, 2022, doi:10.3390/nu14071472_

Round 1

Reviewer 1 Report

In the work entitled ,, Effects of preoperative oral nutritional supplements on Im-2 proving postoperative early enteral feeding intolerance and 3 short-term prognosis for gastric cancer: A prospective, single- 4 center, single-blind, randomized controlled trial "Feng-Jun et al. explain how the dietary preparation of a patient for surgery affects health after surgery.

A large group of patients was involved in the experiment. Their health status was well characterized, and all the features of the patients that could influence the results of the experiment were thoroughly described. The goal of the experiment was also correctly described.

The presentation of the results in the form of a collective table is clear. Graphs enrich your work.

I appreciate that the Authors applied their conclusions directly to the results and sincerely admitted that that short-term preoperative ONS cannot significantly improve FI and the energy compliance rate in the early stage after radical 24 gastrectomy.

I believe that the theoretical introduction to the thesis could be developed. Explain the importance of nutrition in the prevention and supportive treatment of cancer. Perhaps there is literature available confirming that proper supplementation of patients after surgery accelerates recovery.

I encourage the authors to conduct further research. 

Author Response

Dear Reviewer,

Thank you very much for your opinion and recognition. The importance of nutrition in the prevention and supportive treatment of various diseases is gradually recognized, especially in malignant tumors. As you suggested, we are conducting further research and trying to explore the underlying mechanism. Thank you again for your valuable advice!

Kind regards,

Mis Feng-Jun He

E-mail: 1277409022@qq.com

Reviewer 2 Report

The authors evaluated early enteral nutrition after abdominal surgery and also took into account food intolerance.
Do the authors have information on whether certain medicinal plants can alleviate the problem of food intolerance?
Did the authors also consider the possibility of weight gain?
The conclusions are really very concise.
What is the rationale for this nutritional preparation?

Author Response

Dear reviewer,

Thank you again for your questions. For the first question, SimethiconeEmulsion is often used to treat discomfort caused by the intestinal pneumatosis, such as bloating, which may improve intestinal tolerance to some extent, and we are conducting further research.  Then, in our study, there was not much set of observed indicators, such as weight gain or loss, as the reviewer stated, weight loss or gain may be objective evidence to assess the effect of nutritional treatment, but the main focus of this study is is intestinal tolerance. Finally, the rationale for this nutritional preparation is derived from on basic research findings, which has been described in the article. Animal experiments showed that food stimulation has played a regulatory role in the mouse intestinal microecology, while alternately accepting a high-fat, high-sugar diet or a low-fat, high-plant polysaccharide diet can lead to periodically dynamic adjustment in the mouse intestinal microecological system, and each intestinal flora fluctuation caused by specific food reaches a new equilibrium after 3.5 days. Most of the changes in flora characteristics are reversible.

Mind regards,

Wen Zhuang

Reviewer 3 Report

The manuscript entitled "Effects of preoperative oral nutritional supplements on improving postoperative early enteral feeding intolerance and short-term prognosis for gastric cancer: A prospective, single center, single-blind, randomized controlled trial" by Feng-Jun He et al. is report of a single-blind randomized controlled trial that investigated the effect of nutritional supplement versus dietary advice alone (DA) in the preoperative oral nutrition. The study concludes that a short-term, preoperative oral nutritional supplement (ONS) does not improve significantly the feeding intolerance (FI) rate.

The study has been conducted with 66 patients (plus one excluded), of which 32 comprised the experimental and 35 the control group. Although the number of subjects is not sufficient, the research design is appropriate and the statistical analysis of the results is solid. The trial, albeit not of high originality, is comprehensive overall. The results are clearly presented and the conclusion of the study is supported by the analysis of the outcome as a function of the patients' characteristics.

Overall, I see merit in the conclusion of the trial in a way as to promote further research in the field. Therefore, I suggest the authors to elaborate on the future perspectives of similar studies within the Conclusions section. I found only a couple of minor issues for revision.

  1. Minor grammatical issues should be corrected throughout the text. For example: change "ml" to "mL" and "p=" in italics.
  2. Reference [5] is not useful in an international publication. Is there an English version of the article to be cited using DOI?

Author Response

Dear reviewer,

Thank you again for your suggestions. I have revised the article according to your suggestions. As you can see The reference [5] is repeated with the previous references and can be removed. The revisions is attached as follow.

Mind regards,

Wen Zhuang
